# Role of the Furosemide Stress Test in Renal Injury Prognosis

**DOI:** 10.3390/ijms21093086

**Published:** 2020-04-27

**Authors:** Armando Coca, Carmen Aller, Jimmy Reinaldo Sánchez, Ana Lucía Valencia, Elena Bustamante-Munguira, Juan Bustamante-Munguira

**Affiliations:** 1Department of Nephrology, Hospital Clinico Universitario de Valladolid, 47003 Valladolid, Spain; a.coca.rojo@gmail.com (A.C.); mcaller@saludcastillayleon.es (C.A.); jreinaldo@salud.castillayleon.es (J.R.S.); avalenciape@saludcastillayleon.es (A.L.V.); 2Department of Intensive Care Medicine, Hospital Clinico Universitario de Valladolid, 47003 Valladolid, Spain; ebustamante@saludcastillayleon.es; 3Department of Cardiac Surgery, Hospital Clinico Universitario de Valladolid, 47003 Valladolid, Spain

**Keywords:** acute kidney injury, renal biomarkers, furosemide stress test, functional assessment

## Abstract

Risk stratification and accurate patient prognosis are pending issues in the management of patients with kidney disease. The furosemide stress test (FST) has been proposed as a low-cost, fast, safe, and easy-to-perform test to assess tubular integrity, especially when compared to novel plasma and urinary biomarkers. However, the findings regarding its clinical use published so far provide insufficient evidence to recommend the generalized application of the test in daily clinical routine. Dosage, timing, and clinical outcomes of the FST proposed thus far have been significantly different, which further accentuates the need for standardization in the application of the test in order to facilitate the comparison of results between series. This review will summarize published research regarding the usefulness of the FST in different settings, providing the reader some insights about the possible implications of FST in clinical decision-making in patients with kidney disease and the challenges that research will have to address in the near future before widely applying the FST.

## 1. Introduction

Acute kidney injury (AKI) is an intricate clinical syndrome defined by a sudden decrease of kidney function, the accumulation of nitrogen waste products such as urea, electrolyte, and acid-base disturbances and volume overload [1]. The incidence of such complication has been increasing in recent years, affecting 20% of adult and 33% of pediatric patients during hospital admission, especially among subjects with predisposing factors, such as advanced age, diabetes, cardiovascular disease, chronic kidney disease (CKD), or those exposed to nephrotoxins or to cardio-pulmonary bypass. Patients developing AKI have increased morbidity and worse surviving rates than those with normal renal function. In addition, it increases ICU and in-hospital stay, risk of infection, and hospitalization costs. AKI associates a pooled mortality rate of 23.9% in adults and 13.8% in children, rates that increase with higher degrees of severity [1,2,3].

Diagnosis of AKI should comprise several steps, including a thoughtful clinical evaluation, physical examination, consideration of alternative diagnoses, and laboratory data. An abrupt increase of serum creatinine (≥0.3 mg/dL) is still the laboratory finding most closely associated to AKI [4,5,6]. Creatinine is an uncharged 113 Da molecule formed in muscles from creatine, freely filtered at the glomerulus and completely eliminated through the kidney in healthy subjects [7]. Although serum creatinine fulfills most requisites of an ideal filtration marker, it is far from being perfect. In healthy individuals approximately 15% of urinary creatinine is secreted in the proximal tubule, a percentage that can be increased in CKD patients [8]. Additionally, as a product of muscular catabolism, serum creatinine is not an adequate marker of kidney function in subjects with extremely high or low muscle mass such as the elderly or children [9]. Certain drugs (i.e., trimethoprim, cimetidine) can also interfere with tubular secretion of creatinine, producing an increase of serum creatinine levels without real kidney function loss [10,11]. Finally, there is a 48–72 h delay between actual kidney injury and the rise of serum creatinine, which limits early diagnosis and initiation of the appropriate therapeutic measures [12].

Several possible solutions have been developed to overcome this issue. Equations that estimate glomerular filtration rate (eGFR) take into account individual characteristics such as age, gender, weight, or ethnicity to better estimate renal function [13,14]. However, serum creatinine-derived eGFR equations should not be used in the AKI setting due high biases and unacceptably poor performance [15]. Moreover, GFR is not a constant parameter; it changes throughout the day and it is modified by protein consumption and other processes [16].

The development of proteomic technology has triggered extensive research in novel protein indicators that may help characterize AKI mechanisms, improve risk stratification, and facilitate clinical decision making and treatment response monitoring [17]. As a result, multiple potential markers have been discovered in recent years, such as kidney injury molecule-1 (KIM-1) [18,19], neutrophil gelatinase-associated lipocalin (NGAL) [20,21], cystatin C [22], N-acetyl-β-D-glucosaminidase (NAG) [23,24], or liver fatty-acid binding protein (L-FABP) [25,26]. However, despite intensive research effort, none of the discovered biomarkers have managed to replace serum creatinine in clinical practice. Timing of sample procurement, differences in urine concentration and flow, inconsistency among laboratory assays, or higher price are some of the barriers that routine application of novel renal biomarkers must overcome [27]. Additional validation studies that may associate biomarker levels to patient-centered clinical outcomes such as dialysis or death are needed [28].

## 2. Kidney Tubular Stress Test Assessment

Human organ systems have developed the capability to increase their workload in stressful situations. The analysis of this reserve capacity is a useful tool to uncover subclinical disease [16]. Reserve capacity tests are widely applied to study other pathologies, such as coronary artery disease (dobutamine stress echocardiography or exercise electrocardiogram tests). The kidney reserve capacity is built upon two main components, glomerular and tubular (Figure 1). The degree of injury of each component in AKI or CKD can present great variability and be completely independent. Therefore, testing both components could help describe the underlying pathophysiological process with much greater accuracy. 

Glomerular reserve testing has been described in detail, but is sparingly used in day-to-day clinical practice. In brief, GFR, which is commonly used as a surrogate of kidney function, depends on age, sex, weight, or diet and presents great fluctuation among individuals. In healthy subjects, a protein load of 1–1.2 g/kg can induce a considerable increase of GFR above its baseline in 60–120 min. The difference between baseline and maximum GFR is considered the renal glomerular function reserve and is directly associated with stress-associated nephron recruitment and increased renal blood flow [16]. However, the lack of large cohort studies that may help describe the population variability of renal glomerular function reserve is an important limitation of this test. 

The study of tubular reserve capacity is a relatively new diagnostic tool whose clinical application holds great potential [16,29]. The tubules and tubulointerstitium occupy a significant portion of the kidney and are responsible for a wide variety of functions, such as water and electrolyte handling, secretion of endogenous and exogenous acids, and protein catabolism. Tubular epithelial cells (TECs) regulate tubulointerstitial inflammation and are key mediators of tissue repair and fibrosis processes due to their capability to release cytokines, chemokines, and reactive-oxygen species [16,30]. Damaged TECs facilitate tubulointerstitial inflammation due to their capability to modulate the immune response through the formation of several proinflammatory cytokines such as interleukin-6, interleukin-34, or tumor necrosis factor alpha [30,31]. The consequent macrophage infiltration and neutrophil recruitment can aggravate tubular cell injury and perpetuate the pro-inflammatory response to AKI [32,33]. Furthermore, TECs can endure adaptative changes after kidney injury and modify their structure and phenotype, increasing the production of profibrotic factors that stimulate fibroblast proliferation, tubular cell de-differentiation, and epithelial–mesenchymal transition such as connective tissue growth factor, tubular growth factor beta, or renin-angiotensin components [34,35,36]. The epithelial–mesenchymal transition of TECs is a potential key point in the progression of renal fibrosis. This phenomenon involves the replacement of epithelial-type markers, such as E-cadherin or cytokeratin, for mesenchymal-type markers such as vimentin, fibronectin, or type I collagen [30,37].

Recent studies have also described the role of incomplete healing of tubulointerstitial damage as a possible link between AKI and CKD. Although TECs have several repair mechanisms to facilitate full recovery after injury, the repair process can be halted in an intermediate phase, inducing cell atrophy and fibrosis [38]. Mitochondria also play a significant part in AKI associated cell death. The use of SS-31, a drug with mitochondria-protecting effects, has been shown to diminish the development of ischemic damage and interstitial fibrosis in the kidney [39]. Additionally, disturbances of TEC energy-producing pathways, fundamentally fatty acid oxidation, can induce cell death, fibrogenesis, and inflammation [40].

The main tool to assess tubular function is to study tubular secretion of an endogenous or exogenous substance, such as creatinine or furosemide. Salt or acid loading can be used to analyze the efficiency of the tubule to eliminate sodium or acid, while water deprivation or the administration of desmopressin would serve as tests of the concentrating capabilities of the tubule [16].

## 3. Furosemide Stress Test

Furosemide is a short-acting loop diuretic frequently used to treat hypertension, acute or chronic heart failure, or cirrhosis-associated volume overload [41]. This drug can also be used for diagnostic purposes, which constitutes the fundamental bases of the furosemide stress test (FST). The FST is based on the pharmacokinetic properties of furosemide and is aimed to assess the functional integrity of the renal tubule. The drug is strongly bound to plasma proteins and reaches the proximal tubule lumen through active secretion using the human organic acid transporter (hOAT) pathway present in the proximal tubule [42]. After entering the tubule lumen, furosemide blocks the Na^+^-K^+^-2Cl^−^ symporter located in the thick ascending limb of the loop of Henle, preventing Na^+^ reabsorption and increasing the urinary volume excreted. Only the protein-bound fraction of the drug is pharmacodynamically active and as such its effect could be reduced in hypoalbuminemia [42]. In patients with CKD, furosemide produces a lower amount of urine despite its prolonged plasma half-life, due to diminished renal blood flow and reduced tubular secretion [42,43]. Furthermore, in the AKI setting, the accumulation of uremic organic acids could reduce the amount of furosemide reaching the lumen of the tubule due to increased competition at the hOAT site [42]. Additional structural modifications of the proximal tubule during AKI, including upregulation of hOAT1 and hOAT3 or reduced expression of several transporters, such as the Na^+^-K^+^-2Cl^−^ symporter, the epithelial sodium channel or the Na^+^/K^+^ ATPase could further modify the urine output induced by furosemide [44,45].

In this setting, furosemide-induced urinary output has been proposed as a surrogate marker of the integrity of renal tubular function which could help clinicians identify patients with tubular injury and at higher risk of AKI or CKD progression. The clinical utility of the FST has been tested in different backgrounds such as AKI in the critically ill, kidney transplantation, or CKD prognosis (Appendix A) since its inception in 1973 [46]. Due to the growing relevance of AKI in different clinical settings, such as those undergoing cardiac surgery, the potential application of the FST as a tool to better characterize the degree of tubular injury, achieve an earlier diagnosis of AKI and help initiate the most adequate treatment aimed at minimizing AKI-associated morbidity and mortality and preventing AKI-to-CKD transition deserves further attention.

To perform the present review, we searched the PubMed, Web of Science, and Scopus databases to identify relevant published studies in English. Search terms included a combination of subject headings, abstracts, and keywords (e.g., furosemide stress test, furosemide biomarker). Conference papers were excluded from this review. 

## 4. Critical Care

Baek et al., first described in 1973 the application of a furosemide challenge in critically ill, postoperative patients [46]. The study included 38 patients admitted to intensive care units without a past history of CKD. A subset of 15 adequately hydrated patients without diagnosis of AKI and a free water clearance between +15 and -15 received a furosemide bolus dose ranging from 80 to 400 mg. The inability to produce an adequate response to the furosemide challenge was assumed by the authors as a predictor of imminent AKI in this set of patients.

In 2013, Chawla et al., developed a standardized version of the FST [47]. The authors studied two cohorts of 23 and 54 critically ill patients, respectively. All recruited subjects suffered stage I or II AKI according to the Acute Kidney Injury Network (AKIN) classification [5]. Dosing of furosemide was standardized: loop diuretic naïve patients received an intravenous dose of 1 mg/kg while those previously treated with loop diuretics were administered a dose of 1.5 mg/kg. Urine output during the 6 h after furosemide administration was replaced with either saline or Ringers lactate in a 1:1 ratio. The main outcome of the study was the progression to AKIN stage III within 14 days after furosemide administration. The FST was fairly safe, with no adverse events or episodes of hypotension considered attributable to it. A 2-h urine output cutoff of 200 cc showed the best combination of sensitivity (87.1%) and specificity (84.1%) and was a robust predictor of progression to AKIN stage III. The area under the receiver operator characteristic curves (AUC) for the complete urine output over the first 2 h after the FST to predict the primary outcome was 0.87. Nonetheless, the authors highlight that, for the test to be applied, patients should be euvolemic and any obstruction to urinary flow should have been resolved before the administration of the FST.

A secondary analysis of the same cohort was published in 2017 [48]. In this study, Koyner et al., compared the predictive capacity of FST with that of eight plasma and urinary biomarkers such as NGAL, KIM-1, interleukin-18, uromodulin, tissue inhibitor of metalloproteinases (TIMP-2), IGF-binding protein-7 (IGFBP-7), or albumin-to-creatinine ratio. The 2-h urine output after FST outperformed all studied urinary biomarkers when predicting progression to AKIN stage III with an AUC of 0.87. Furthermore, the FST outperformed most urinary biomarkers when predicting the need of renal replacement therapy (RRT) (AUC: 0.86) or a composite outcome of patient death or progression to AKIN stage III (AUC: 0.81). The combined use of FST and urinary biomarkers to predict outcomes did not significantly improve the performance of FST as a predictor when used alone. Authors concluded that FST was a promising tool that may help clinicians to improve risk stratification in patients with early stages of AKI.

Another approach to test the predictive capacity of the FST was used by van der Voort et al. [49]. In this study, urinary production was measured during a 4-h period after termination of continuous renal replacement therapy (CRRT) in a sample of critically ill patients with AKI. After this period, a subset of patients received either furosemide 0.5 mg/kg/h or placebo with a 4-h urine output repeated measurement after 24 h. In this study, both spontaneous urinary production after CRRT cessation and furosemide-induced urine output were significantly higher in those subjects with immediate recovery of renal function. The AUC for the total urinary output over the first 4 h after the FST to predict in-hospital renal recovery was 0.79. The authors postulated that the FST could be used as a potential predictor to assess renal function recovery after CRRT.

In 2018, Matsuura et al., retrospectively analyzed 95 patients admitted to an intensive care unit and who were treated with bolus furosemide [50]. Authors excluded those patients with AKI stage 3 according to the Kidney Disease: Improving Global Outcomes (KDIGO) AKI classification and those who received a continuous intravenous furosemide infusion. The final sample included 95 subjects with either no AKI or AKI stage 1 or 2. Furosemide responsiveness was defined as the urine output (ml) produced in 2 h divided by the dose of furosemide administered (mg). Urinary biomarkers such as plasma NGAL and urinary L-FABP and NAG were also determined. Main outcomes were progression to AKI stage 3 and a combined outcome of progression to AKI stage 3 or patient death. Furosemide responsiveness was significantly higher among non-progressors, with an AUC for the combined outcome of 0.88, which was higher than that of plasma NGAL (0.81), urinary L-FABP (0.62), or urinary NAG (0.53). Furthermore, the efficacy of furosemide responsiveness was tested in a group of 51 patients with plasma NGAL levels >142 ng/mL at the time of furosemide administration. In this subset, furosemide responsiveness presented an AUC of 0.88 to predict the composite outcome AKI stage 3 progression or patient death.

An alternative clinical use for the FST was tested by Lumlertgul et al., in a prospective, multicenter, randomized controlled trial [51]. In this study, investigators used the FST as an initial triage strategy to identify patients for randomization to different RRT initiation times. Those subjects with poor response after the FST were randomized to either early or standard RRT initiation. Although in this study FST was not assessed as a predictor of clinical outcomes, the test proved to be a safe and effective tool to stratify AKI patients at high risk for RRT.

Recently, Rewa et al., prospectively analyzed the predictive power of FST in a sample of 92 critically ill patients with AKIN stage I or II, recruited from five intensive care units [52]. Patients with evidence of volume depletion, active bleeding, or obstructive uropathy were excluded from the analysis. The dose of intravenous bolus furosemide administered was that proposed by Chawla et al.: 1 mg/kg for loop diuretic naïve patients and 1.5 mg/kg for those previously treated with loop diuretics. The primary outcome was progression to AKIN stage III within 30 days of FST administration. FST-induced urine output was a significant predictor of the primary outcome, with an AUC of 0.87. However, the FST failed to predict in-hospital patient survival. Moreover, in a multivariate logistic regression model that included the Acute Physiology and Chronic Health Evaluation (APACHE II) score, baseline urine output or serum creatinine at the time of FST, only FST-induced urine output and sex were significant predictors of AKI progression. Authors also registered adverse events that occurred after FST; 9.8% of patients suffered an episode of clinically significant hypotension and 5.4% developed hypokalemia or hypomagenesemia, with no life-threatening events recorded. Authors concluded that FST was a safe and effective predictive tool in patients with mild to moderate AKI.

Sakhuja et al., examined if FST could be used to detect AKI stage 3 patients at risk of needing RRT [53]. Due to the retrospective nature of this study, the furosemide dose was not standardized, but only subjects that received at least 1 mg/kg of intravenous bolus furosemide or its equivalent dose of intravenous bumetanide were included. Patients that had previously received loop diuretics before the FST were excluded from this analysis. Primary and secondary outcomes for Sakhuja et al., were defined as the need for urgent dialysis within 24 or 72 h after the FST. A total of 687 patients were included in the final sample. Dialysis had to be administered to 162 patients (23.6%) during the first 24 h after FST. The 6-h urinary production after FST had only modest discriminative power to predict need of dialysis within the next 24 h, but, according to authors, its application could be useful to evaluate the need for dialysis in critically ill patients with AKI stage 3.

Finally, the usefulness of furosemide response as a predictor of AKI has also been tested in the pediatric critical care setting. Borasino et al., retrospectively examined a sample of 90 infants and neonates younger than 90 days old who received at least one dose of furosemide in the first 24 h after cardiopulmonary bypass surgery [54]. Average furosemide dose was 1.1 ± 0.3 mg/kg. The primary endpoint of the study was the development of cardiac surgery-associated AKI, defined as the doubling of serum creatinine within 72 h of index surgery or a urinary output <0.5 mL/kg/h on average in a 24 h period over the first 72 h after index surgery. Response to furosemide predicted cardiac surgery-associated AKI in this setting, with an AUC of 0.69. Additionally, furosemide response predicted peritoneal dialysis initiation and fluid overload.

## 5. Kidney Transplantation

There is a paucity of literature regarding the application of FST outside of the intensive care setting. The FST has been tested as a predictive tool after kidney transplantation with different approaches regarding timing of administration and furosemide dosage. McMahon et al. [55] published in 2018 a single-center retrospective analysis of a random sample of 200 deceased-donor kidney transplant recipients who received an intraoperative bolus of 100 mg furosemide. Urinary production was measured 2 and 6 h after furosemide administration. The primary outcome was the development of delayed graft function, defined as the need of dialysis within 7 days of transplantation. Authors included as secondary outcomes safety endpoints such as incidence of hypotension or hypokalemia during the first 24 h after the bolus, graft loss, rejection, death with functioning graft or length of hospital stay. Subjects who developed delayed graft function presented a significantly lower urine output 2 and 6 h after FST. A 6-h urinary output <600 mL (which defined FST non-responders) presented an AUC of 0.85 for the development of DGF. Regarding safety-related outcomes, no episodes of hypotension (defined as mean arterial pressure <60 mmHg) or change in plasma potassium levels were observed in the sample. Although FST non-responders showed longer length of in-hospital stay, the rates of graft loss or death were similar between both groups.

The usefulness of FST after kidney transplantation was also studied by Udomkarnjananun et al. [56]. The authors prospectively studied a sample of 59 adult deceased-donor kidney transplant recipients without hypoalbuminemia or surgical complications that required reoperation during the first 24 h after transplantation. Dry weight adjustment was used to ensure euvolemia prior to transplantation. An intravenous bolus dose of 1.5 mg/kg furosemide was administered 3 h after allograft reperfusion. Urinary production was registered for 6 h after the bolus. Each mL of urine produced during the first 24 h was replaced by 1 mL of saline to avoid volume depletion. The primary outcome was incidence of DGF, using the same definition applied by McMahon et al. Mean cumulative urine volume was significantly lower in those who developed DGF compared to that of non-DGF patients. A 4-h cumulative urine output <350 mL presented the highest accuracy to predict DGF with an AUC of 0.94. The FST was the only significant predictor of DGF in multivariate logistic regression analysis. Moreover, FST was the most accurate predictor of DGF when compared to urinary NGAL, resistive index of renal arteries measured by ultrasonography or effective renal plasma flow measured by 99mTc-MAG3 renography.

These two single-center studies show that FST could improve risk stratification in the early post-transplant period, promptly detecting patients with higher risk of DGF who could benefit from an early initiation of therapeutic interventions. FST was a more accurate predictor of DGF than oliguria (defined as a daily urinary production <400 mL) [55], ultrasonography, 99mTc-MAG3 renography or novel biomarkers such as urinary NGAL. The test was also safe and well tolerated, in line with published results in critically ill patients.

However, although these studies provide an interesting starting point for the application of FST as a predictive tool after kidney transplantation they also suffer significant limitations: both were small sized, single-center studies, limiting external validity of results. The analysis by McMahon et al., was a retrospective review, which means exposure or outcome assessment could not be controlled. Furthermore, furosemide dosage had different timing and dosage in both studies. The possibility of volume depletion after transplant surgery and/or obstructive uropathy due to urinary elimination of blood clots, complications which are commonly associated to active bleeding, were not adequately addressed in these studies. Therefore, additional prospective multi-center studies should be planned to analyze if FST could improve risk stratification and patient management after kidney transplantation.

## 6. Other Clinical Settings (CKD)

As previously stated, most research regarding the use of FST is centered in the AKI setting. However, Rivero et al. [57] have recently published a prospective study regarding the usefulness of FST as a tool to assess interstitial fibrosis in a sample of CKD patients. To that end, the authors included adult subjects admitted for a kidney biopsy, including transplant recipients. Hypovolemic or subjects with hemodynamic instability were excluded from the study. A standardized dose of 1 mg/kg furosemide, or 1.5 mg/kg furosemide if exposed to loop diuretics during the seven days prior to FST was administered. Fluid therapy was dispensed according to post-FST urine output to avoid furosemide-induced volume depletion. A nephropathologist assessed kidney interstitial fibrosis percentage using morphometry and classified patients in one of three categories: <25%, 26–50% and >50%. Subjects with >50% interstitial fibrosis had a significantly lower urine output after FST, with an inverse correlation between FST response and degree of fibrosis. FST could thus be a potential tool to non-invasively assess interstitial fibrosis, offering a complementary instrument to eGFR and proteinuria to evaluate prognosis and disease progression in CKD patients.

## 7. Current Limitations and Future Challenges

The FST is a safe, non-invasive, easy to perform, low-cost tool to evaluate the severity of tubular injury which has shown promising initial results in different clinical contexts. Its capacity as a predictive tool could facilitate risk stratification and clinical decision-making in areas as disparate as AKI, kidney transplantation, or CKD. However, there is still a long way to go before the FST can be included in diagnostic and treatment algorithms. Most published clinical research to date is based on small-sized, single center pilot or feasibility studies. Several studies are based on retrospective analysis of patients who received a furosemide bolus, which makes the task of controlling possible sources of bias extremely difficult. Moreover, doses and timing of furosemide administration are highly variable, even in studies framed within the same clinical setting. Standardization of dosage, such as that proposed by Chawla et al. [47] should help to homogenize the FST in order to facilitate comparison of results between studies.

Additionally, some of the published studies solely rely on AUC values to define the predictive capacity of the test. It has been pointed that such statistic may not be the most adequate choice to assess models that predict risk or stratify individuals into risk categories, a setting in which calibration may play a significant role [58]. In these instances, actual or absolute predicted risk, which is not captured by the AUC, could be of outmost interest. Therefore, when comparing models for risk prediction, a combined analysis including global model fit and analysis of calibration and discrimination would be recommended.

## 8. Conclusions

In summary, tubular stress test assessment using the FST is a rediscovered and promising new tool to stratify risk in different kidney diseases. Multi-center, prospective studies with large enough sample sizes, applying standardized furosemide dosage and timing and comparing the FST to other novel plasma and urinary biomarkers are necessary to adequately validate results and define the possible clinical applications of the test.

## Figures and Tables

**Figure 1 ijms-21-03086-f001:**
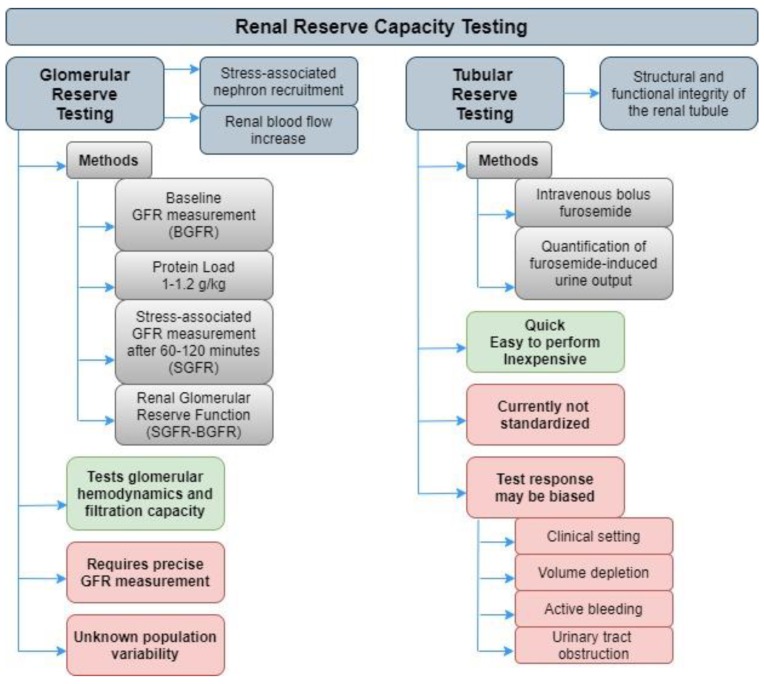
Renal reserve capacity testing.

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
