# Peer review of "Role of the Furosemide Stress Test in Renal Injury Prognosis"

_ijms, 2020, doi:10.3390/ijms21093086_

Round 1

Reviewer 1 Report

This is a well researched and written review of the use of the furosemide stress test in patients with acute kidney injury.  It outlines the history, physiology and pharmacology behind the test and also the limitations of the test.  The manuscript is well referenced and the single figure is instructive and easy to read.

Author Response

We thank the reviewer for their comments, several language and grammar changes have been introduced throughout the text to facilitate it reading and comprehension.

Reviewer 2 Report

Thank you for giving me the opportunity to review this paper focusing on the utility of furosemide test in AKI. The furosemide stress test (FST) is feasible and well tolerated even in critically ill patients with early AKI. It has been proposed as a novel /additional assessment of tubular function with predictive capacity to identify those patients with severe and progressive AKI. However, it has not been introduced as a routine in clinical practice, as its accuracy and validation are still under investigation.

The paper is a detailed and comprehensive manuscript which provide detailed data on these issues.

 I have only some comments:

  1. Did the authors follow a certain methodology to make sure they include all the papers concerning furosemide test in their study (key word used, search engines etc)? Please clarify.
  2. Could the authors provide a table with the results of the analyzed studies? This will be really helpful for the reader.

Author Response

 I have only some comments:

  • Did the authors follow a certain methodology to make sure they include all the papers concerning furosemide test in their study (key word used, search engines etc)? Please clarify.

Response

Thank you very much for your comment, additional information on search terms and databases used in this review has been included at the end of section 3. Due to the inclusion of a third database in our search (WoS), two additional studies have been added to the review.

Please see table 1.

Please see lines 147-150; 206-210 and 237-245

  • Could the authors provide a table with the results of the analyzed studies? This will be really helpful for the reader.

Response

We agree with the reviewer, Table 1 has been updated to summarize results of all analyzed studies on the predictive capabilities of furosemide administration.

Please, see table 1.

Several language and grammar changes have been introduced throughout the text to facilitate it reading and comprehension.